# Rapid On-Farm Testing of Resistance in *Lolium rigidum* to Key Pre- and Post-Emergence Herbicides

**DOI:** 10.3390/plants10091879

**Published:** 2021-09-10

**Authors:** Martina Badano Perez, Hugh J. Beckie, Gregory R. Cawthray, Danica E. Goggin, Roberto Busi

**Affiliations:** 1Australian Herbicide Resistance Initiative, School of Agriculture and Environment, The University of Western Australia, Perth 6009, Australia; martinabadano@hotmail.com (M.B.P.); danica.goggin@uwa.edu.au (D.E.G.); roberto.busi@uwa.edu.au (R.B.); 2School of Biological Sciences, The University of Western Australia, Perth 6009, Australia; gregory.cawthray@uwa.edu.au

**Keywords:** annual ryegrass, clethodim, epidemiology, glyphosate, herbicide bioassay, herbicide resistance, pyroxasulfone, trifluralin

## Abstract

Overreliance on herbicides for weed control is conducive to the evolution of herbicide resistance. *Lolium rigidum* (annual ryegrass) is a species that is prone to evolve resistance to a wide range of herbicide modes of action. Rapid detection of herbicide-resistant weed populations in the field can aid farmers to optimize the use of effective herbicides for their control. The feasibility and utility of a rapid 7-d agar-based assay to reliably detect *L. rigidum* resistant to key pre- and post-emergence herbicides including clethodim, glyphosate, pyroxasulfone and trifluralin were investigated in three phases: correlation with traditional pot-based dose-response assays, effect of seed dormancy, and stability of herbicides in agar. Easy-to-interpret results were obtained using non-dormant seeds from susceptible and resistant populations, and resistance was detected similarly as pot-based assays. However, the test is not suitable for trifluralin because of instability in agar as measured over a 10-d period, as well as freshly-harvested seeds due to primary dormancy. This study demonstrates the utility of a portable and rapid assay that allows for on-farm testing of clethodim, glyphosate, and pyroxasulfone resistance in *L. rigidum*, thereby aiding the identification and implementation of effective herbicide control options.

## 1. Introduction

Since the 1880s, highly adaptable *Lolium rigidum* Gaud. (annual or rigid ryegrass) accessions were introduced into southern Australia as a pasture for livestock grazing [1]. After a substantial shift towards annual cropping in the 1950s, the control of *L. rigidum* infestations became a priority in cereals, oilseeds, and pulses because of the highly competitive and economically damaging nature of the weed even at early developmental stages [2]. Weeds such as *L. rigidum* are a major constraint for Australian farming systems; the financial cost from yield loss and weed control expenditure amounts to AU $3.3 billion per year [3]. Effective control of *L. rigidum* relies on systematic herbicide use as a key component of crop protection in Australian minimum-tillage dryland farming systems, enabling growers to sow crops in fields with stubble retention and avoid erosive and labour-intensive soil cultivation [4]. However, this high reliance on herbicides to control weeds promotes the evolution of herbicide-resistant genotypes [5]. The most challenging species is *L. rigidum* due to (1) its propensity to rapidly evolve resistance to a wide range of herbicides with different modes of action [6,7,8] and (2) highly variable seed dormancy levels, allowing different cohorts to germinate across the growing season and avoid pre-seeding knockdown (burndown) and/or in-crop herbicide application [9].

The detection of resistant weed populations generally occurs in the field as a consequence of the failure of a herbicide treatment (<80% efficacy) that was previously effective. If resistance could be detected earlier by proactively scouting fields and routinely screening weed populations, it may be possible to better anticipate the causes of herbicide resistance and thereby implement changes in management designed to achieve effective weed control and minimize the spread of resistance [10,11]. However, proactive resistance testing is not a commonly adopted practice among Australian growers because of the time required for seed collection in fields, uncertainty over collection methods, or cost (R. Busi, unpublished survey data). Testing seed samples for resistance using traditional pot-based methods is expensive, time-consuming, and requires a large amount of space [12]. To encourage growers to test for resistance more frequently, it would be advantageous to provide them or their service providers (e.g., grower groups, agronomists, consultants) with step-by-step guidelines from seed collection to seed testing [13]. Even more important is to develop a faster and less expensive diagnostic technique with a simple interpretation of the results. Methods involving germination of seeds or pollen on herbicide-containing media, incubation of leaf samples in herbicide solutions, spraying of separated grass weed tillers or measurement of hydroponically-grown seedling development in the presence of herbicide have been used in research laboratories [14], but none have been developed for use by growers or service providers themselves to enable on-farm resistance testing.

Accordingly, the aim of this study was to develop and evaluate an easy-to-interpret on-farm test to rapidly detect resistance in *L. rigidum* populations to key pre- and post-emergence herbicides used in Australian grain cropping: clethodim, an acetyl-CoA carboxylase inhibitor; glyphosate, a 5-enolpyruvylshikimate-3-phosphate synthase inhibitor; pyroxasulfone, a very long chain fatty acid elongase inhibitor; and trifluralin, a cell division inhibitor. The strategy to achieve this objective was the preparation of herbicide-impregnated agar in containers in the laboratory, which could be packaged into test kits and distributed to end-users. The test could then be performed by placing putative resistant seeds on the herbicide-containing agar and observing and rating seedling growth after 1 week. To address the feasibility of this goal, we determined (1) how well results of the rapid test correlate with classical pot-based dose-response assays; (2) the effect of seed dormancy as a barrier to the success of the test [10]; and (3) the stability of herbicides in agar to assess the potential portability and shelf-life of the test.

## 2. Results

### 2.1. Development of the Agar Test and Comparison with the Pot Test

It was possible to visually discriminate between standard susceptible (S) and resistant (R) *L. rigidum* individuals in agar-based assays in response to each of the four herbicides (Figure 1). The discriminating range of concentrations for each herbicide that caused 100% mortality and full inhibition of coleoptile elongation in the S population, but ≥2% survival and growth in the R populations were identified. Clethodim at 0.25 to 1μM resulted in 4% to 52% survival in the clethodim-resistant populations (Figure 2A). Glyphosate applied at 400 to 600 μM resulted in a survival response ranging between 6% and 59% for the glyphosate-resistant populations (Figure 2B). For pyroxasulfone, the discriminating concentrations (0.075–0.15 μM) resulted in 2% to 20% survival of the pyroxasulfone-resistant populations (Figure 2C), whereas survival to trifluralin applied from 10 to 50 μM was relatively high in both trifluralin-resistant populations, ranging between 10% and 60% (Figure 2D). The dose-response experiment in pots showed the expected discrimination between the S and R populations, with the former having 100% mortality at the discriminating rate of each herbicide (data not shown). Herbicide-resistant populations, classified according to statistical analysis of LD_50_ values and resistance indices (RI), were accurately identified using the agar whole plant test (100% correlation with pot assay) (Table 1). Even though RI values varied between the two tests, a value ≥2 is universally considered as being resistant, whether a herbicide-resistant weed or crop.

### 2.2. Validation of the Agar Test with Field Populations

Agar and traditional pot experiments were conducted to compare the survival response of a large number (100) of field populations collected from the Western Australian wheat belt. As was expected from previous surveys [15,16], the overall frequency of resistance to trifluralin and clethodim was >40% of populations at the discriminating dose used in agar and pots. There were only five populations resistant to glyphosate and no populations resistant to pyroxasulfone (confirmed in both assays). The data for clethodim and trifluralin showed highly significant correlations between percent survival in agar-based vs. pot-based resistance tests (non-significant for glyphosate and pyroxasulfone due to low frequency of resistant samples; Table 2). In agreement with the results described in the previous section, the agar test classified S (0% survival) and R populations (≥2% survival) similarly to the pot test, although specific percent survival rates may have varied.

### 2.3. Interaction between Seed Dormancy and the Agar Test

In all experiments, the responses of seeds germinated on the laboratory window sill were similar to those germinated in the growth cabinet (data not shown); therefore, only the latter results are presented. Several methods of chemical or mechanical scarification or chemical stimulation (Appendix A) were tested on seeds with primary or conditional dormancy to determine if the requirements for dark stratification and alternating light and temperature could be overcome. In the absence of any treatment, germination of conditionally dormant seeds after 7 d was 50%, while only 5% of the seeds with primary dormancy had germinated during this time (Appendix A). No scarification or stimulation method was successful in increasing germination to >70% in the first 7 d (overall average of 28%).

The germination of highly dormant seeds placed directly under non-stratified germination conditions in the presence or absence of herbicide was too low after 7 d at 25/15 °C with a 12 h photoperiod for the results to be reliable; however, the glyphosate-resistant population, which had low dormancy, did exhibit a glyphosate-resistant phenotype (Figure 3A). Dark stratification of seeds in the absence of herbicide for 21 d improved the germination percentage of the dormant populations but resulted in premature germination of the less-dormant populations so that fewer ungerminated seeds were available for the 7 d germination test. Based on seed germination after 7 d, all four of the herbicide-resistant populations appeared to be glyphosate-resistant, while the clethodim-resistant population appeared susceptible to this herbicide (Figure 3B).

Dark stratification of seeds in the presence of 0.1× of the discriminating dose of herbicide resulted in apparent resistance to three or more herbicides in all populations except for the non-dormant S control, which only appeared resistant to pyroxasulfone (Figure 3C). In the presence of 0.25× or 1× of the discriminating dose, all dark-stratified populations except the non-dormant S control appeared resistant to glyphosate (Figure 3D,E). Additionally, the clethodim- and pyroxasulfone-resistant populations appeared resistant to trifluralin, and the trifluralin-resistant population appeared resistant to clethodim at 0.25× of the discriminating dose (Figure 3D). The pyroxasulfone- and trifluralin-resistant populations appeared susceptible to these herbicides at 1× of the discriminating dose (Figure 3E).

### 2.4. Stability of Herbicides in Stock Solutions and Agar Plates

There was no degradation of the 10 mM pyroxasulfone and glyphosate stocks stored at 4 °C across the 236 days of the experiment, but some slight (~15%) loss of trifluralin and clethodim was observed (data not shown). When incorporated in agar, clethodim remained stable if stored in the dark at room temperature, but 50% to 70% was lost between 5 and 10 d in the other two treatments (Figure 4). Glyphosate and pyroxasulfone remained stable in agar under all three storage conditions (Figure 4). In contrast, there was an almost complete loss of trifluralin in the first 5 d in plates stored at room temperature in the light or dark, although a much smaller decrease (around 25%) in the samples stored at 4 °C (Figure 4).

## 3. Discussion

The aim of this study was to find a simplified yet accurate supplement to traditional pot-based assays for herbicide resistance screening of *L. rigidum* to key pre- and post-emergence herbicides that could be used on-farm. Using an agar-based test prepared in the laboratory, it would be distributed to growers or their service providers to detect herbicide resistance in the field by simply placing weed seeds into a container and observing coleoptile elongation after a week. *Lolium rigidum* was identified as a suitable model weed species due to its reported capacity to evolve resistance to multiple herbicide modes of action [7] and its importance and prevalence in Australian cropping systems.

Under experimental conditions in the laboratory, the agar test was able to discriminate characterized (non-dormant) populations of *L. rigidum* resistant to herbicides from four different modes of action: clethodim, an acetyl-CoA carboxylase inhibitor; glyphosate, a 5-enolpyruvylshikimate-3-phosphate synthase inhibitor; pyroxasulfone, a very long chain fatty acid elongase inhibitor; and trifluralin, a cell division inhibitor. The agar test was also able to discriminate previously uncharacterized susceptible and resistant field populations of *L. rigidum* to clethodim, glyphosate, and trifluralin (no pyroxasulfone-resistant populations). Although the pot assay better estimates the frequency of resistance (% survival) in a population and the resistance level (resistance index), the agar assay meets the key criterion of accurately identifying susceptible and resistant populations (developing resistance or resistant) of *L. rigidum*.

This study shows that the agar assay requires non-dormant (after-ripened) seeds of *L. rigidum*, which is a probable constraint to the adoption of the test as a portable kit for growers to use on-farm at or near harvest time (typically October-November in Western Australia). However, testing after-ripened weed seeds post-harvest and before crop planting (typically late April-May) should alleviate seed dormancy and, therefore, this constraint. Seeds that do not germinate due to dormancy, or germinate in the last few days of the test, could be interpreted as being susceptible and thus give false-negative results [17]. In the current study, none of the dormancy-breaking treatments tested on dormant *L. rigidum* seeds were able to short-circuit the requirement for weeks of dry after-ripening or days of dark stratification. Dark stratification itself had a tendency to confound the results, with most populations appearing to be glyphosate-resistant when previous pot tests had shown that they were not glyphosate-resistant. Moreover, even the populations previously characterized as pyroxasulfone- or trifluralin-resistant appeared susceptible after dark stratification on these herbicides at the full discriminating dose, while lower-than-discriminating doses resulted in false positives. Further work is needed to fully understand the relative ability of dormant seeds to survive discriminating herbicide doses while undergoing a dormancy release via dark stratification and whether the dormancy release itself is affected by herbicides. The requirement for low-dormancy seeds to ensure accurate results means that freshly collected seeds intended for an on-farm test need to have their dormancy level assessed and then be left to after-ripen if necessary. Even soil pot assays for herbicide resistance screening require seeds with sufficiently high germination [10].

Another limitation of the agar test as an on-farm resistance assay is the rapid loss of trifluralin upon incorporation in agar. With the necessity of using trifluralin-containing agar almost immediately, the shelf-life of an agar test kit would not be sufficient for the time required for the kit to be distributed and utilized by growers or their service providers. Interestingly, the growth of trifluralin-susceptible seedlings was still inhibited after 42 d incubation on trifluralin-containing agar (data not shown), even when trifluralin in agar was depleted after 5 d (Figure 4). This result could be explained by the binding of trifluralin to the seeds, maintaining a high local concentration even when the herbicide evaporates from the surrounding agar. Dinitroaniline herbicides are known to bind strongly to organic matter [18,19], particularly lipids, and therefore are able to accumulate in the cell membrane [20]. Clethodim stability in agar necessitates agar storage at ambient room temperature under dark (e.g., drawer or cabinet) rather than light conditions. However, this qualification should not be a problem for test users.

In summary, the rapid agar test described in this study is a potential on-farm tool for growers or retail agronomists for checking *L. rigidum* populations for resistance to clethodim, glyphosate or pyroxasulfone, which are widely and extensively used in Australian grain cropping. The instability of trifluralin in agar precludes testing for resistance to this pre-emergence herbicide on-farm. This portable, rapid on-farm test can complement pot-based assays typically used by herbicide testing organizations by providing an inexpensive initial or earlier assessment of resistance to these herbicides in a simple, easy-to-use and convenient test kit. This quick test can thereby facilitate greater adoption of herbicide resistance testing by growers or land managers through awareness of the current resistance status of *L. rigidum* populations in their fields, which is an essential prerequisite for deploying effective herbicide treatments.

## 4. Materials and Methods

### 4.1. Plant Material

Well-characterized *L. rigidum* populations with little to no dormancy (>96% germination within 7 d) were used to optimize the agar-based resistance test and assess its correlation to traditional whole-plant pot-based experiments (Table 3). These populations have been characterized as S or R in standard pot-based dose-response experiments using clethodim, glyphosate, pyroxasulfone, and trifluralin. An additional 100 populations collected from different fields in Western Australia in 2018 and 2019 [21] were used to validate the correlation between the agar test and the whole-plant test in pots. Before testing, seed dormancy in these populations was relieved by hydrated storage in the dark at room temperature for 14 d [22]. Populations selected for low (conditional) dormancy requiring alternating light and temperature to trigger germination or very high primary dormancy requiring extensive after-ripening or dark stratification to sensitize them to germination cues [22] were used as controls in experiments to investigate the effect of seed dormancy on agar test results [23] (Table 3).

### 4.2. Optimization of the Agar Test

A commercial formulation of herbicide was added to liquid 0.6% (*w*/*v*) agar and poured to a depth of 1 cm in 5-centimetre deep plastic containers of 12.5 cm diameter. The herbicide concentrations chosen for dose-response experiments were based around discriminating doses identified in preliminary experiments and in the literature [10,12,30]. Clethodim (Sequence, 240 g a.i. L^−1^; Nufarm, Laverton North, Victoria, Australia) was used at concentrations of 0.125, 0.25, 0.5, 1, 2, 3, 4, and 5 μM; glyphosate (Weedmaster ARGO, 540 g a.i. L^−1^; Nufarm, Laverton North, Victoria, Australia) at 10, 100, 200, 300, 400, 500, 600, and 1000 μM; pyroxasulfone (Sakura 850 WG, 850 g a.i. kg^−1^; Bayer Crop Science, Sydney, New South Wales, Australia) at 37.5, 75, 150, 300, 600, and 1200 nM; and trifluralin (Triflur X, 480 g a.i. L^−1^; Nufarm, Laverton North, Victoria, Australia) at 5, 10, 20, 25, 30, 50, and 100 μM. Herbicide-free controls were included in every experiment. Twenty-five seeds per treatment were placed on the surface of the herbicide-impregnated agar and incubated in a growth chamber at 25/15 °C day/night with a 12 h photoperiod of cool white fluorescent light (90 μmol m^−2^ s^−1^) [23] for 7 d. Individual seedlings were considered resistant if they actively grew, with the length of their coleoptiles reaching ≥ 4 cm. The agar container was the experimental unit. There were three replicates of each herbicide concentration, and the experiment was repeated. The minimum dose that killed 100% of the individuals in the S control populations was considered to be the discriminating dose used in the validation experiment (Section 4.4).

### 4.3. Pot-Based Dose-Response Experiments

The results of the agar test were compared with those of a traditional pot-based resistance test by conducting dose-response experiments in 6 cm × 6 cm × 6 cm plastic cells filled with potting mix soil (25% peat moss, 25% washed river sand, and 50% mulched pine bark). For the pre-emergence herbicides (pyroxasulfone and trifluralin), 50 seeds per cell were sprayed while situated on the soil surface and then covered with a thin layer of soil. For the post-emergence herbicides (clethodim and glyphosate), 50 seedlings in each cell were sprayed at the two- to three-leaf stage. Herbicides were applied using a cabinet track sprayer with TeeJet XR11001 flat-fan nozzles (Spraying Systems Co., Wheaton, IL, USA), delivering 106 L ha^−1^ per pass at 200 kPa pressure [8]. The range of doses applied was based around the recommended (1×) field rate for each herbicide (in g a.i. ha^−1^: clethodim, 120; glyphosate, 540; pyroxasulfone, 100; trifluralin, 960) or on the discriminating dose used in published studies [8,31,32,33,34]. Clethodim was applied at 62, 125, 250, 500, and 750 g ha^−1^; glyphosate at 270, 540, 810, 1080, and 1350 g ha^−1^; pyroxasulfone at 12, 25, 50, 100, 150, and 200 g ha^−1^; and trifluralin at 60, 120, 240, 480, and 720 g ha^−1^. Unsprayed controls were also included. Plants were grown in the glasshouse in autumn (March/April) with regular watering and fertiliser application, and survival was assessed at 21 d after the spray application. Those plants that could establish, develop two new leaves and actively grow were counted as survivors. Each cell of 50 individuals was the experimental unit. There were three replicates of each treatment, and each herbicide experiment was performed twice.

### 4.4. Validation of the Agar Test with Uncharacterized Populations

One hundred uncharacterized populations collected from the field in 2018/19 were incubated on agar containing the discriminating herbicide doses identified in the optimization phase of the study. To account for any seed dormancy remaining after 14 d dark stratification, sub-samples were germinated on control agar lacking herbicide. The seedling growth and survival of treated seeds were expressed as a percentage of the untreated control. The populations were also sprayed with the discriminating dose of each herbicide in pot-based tests to determine the correlation between observed resistance levels (% survival) in the agar and pot assays. Plant survival, defined as the percentage of seedlings that had produced at least two new leaves after herbicide exposure, was assessed at 21 d after treatment. For both the replicated agar and pot experiments, a treatment replicate comprised 25 seeds.

### 4.5. Investigation of Interference from Seed Dormancy

To determine if dormant *L. rigidum* seeds can be forced to germinate within the 7 d time frame of the agar test, the well-characterized low- and high-dormancy populations (ND4 and VD4) from Goggin et al. [23] were subjected to a range of potential dormancy-breaking treatments (see Appendix A for full details and references). Seeds were then incubated on agar under either controlled conditions (25/15 °C, 12 h photoperiod) or next to the laboratory window to simulate conditions in a farmer’s house. Germination was counted every 7 d for 42 d, and dead or empty seeds were excluded from calculations. The replicated experiment had three replicates per treatment.

The effect of dark stratification in the presence of herbicides was assessed using population S (Table 3) as a susceptible, non-dormant control; population VD4 as a susceptible, dormant control; and freshly-harvested (not after-ripened) samples from five populations with known resistance to clethodim (population CR), glyphosate (GR), pyroxasulfone (PR), or trifluralin (TR) (Table 3). Seeds (10 per treatment) were imbibed on agar as previously described and placed under environmental conditions of 25/15 °C in a growth cabinet or on the laboratory window sill for 42 d, or first dark stratified at 20 °C for 21 d (dishes were wrapped in foil) before being transferred to germination conditions for a further 21 d. Seeds placed directly under germination conditions were imbibed on a discriminating dose of each herbicide (2 μM clethodim, 400 μM glyphosate, 0.15 μM pyroxasulfone, or 20 μM trifluralin). Dark stratified seeds were incubated on agar containing 0, 0.1×, 0.25×, or 1× of the discriminating dose of each herbicide. Seeds stratified in the absence of herbicide were transferred to agar containing a discriminating herbicide dose at the same time as they were exposed to germination conditions, while seeds stratified on herbicide-containing agar were kept on the same agar during the germination phase. The sub-lethal herbicide treatments were included because of the longer incubation time of the seeds with the herbicides. Germination was recorded every 7 d. For the purposes of this experiment, which assessed herbicide resistance as well as seed germination, a coleoptile length of ≥4 cm was required for a seed to be counted as germinated. There were three replicates per treatment, and the experiment was repeated.

### 4.6. Herbicide Stability in Agar

Technical-grade herbicides (Sigma-Aldrich, Sydney, New South Wales, Australia) were used for the initial optimization of high-performance liquid chromatography (HPLC) methods. Stock solutions of 1 mM clethodim, pyroxasulfone, or trifluralin in 50% acetonitrile were prepared, and dilutions then made to 10–200 µM, 5–10 μL of which was directly injected onto a Nova-Pak C_18_ column (150 mm × 3.9 mm i.d.) with a 4 μm particle size (Waters, Milford, MA, USA) attached to a 600E dual-head pump with 717 Plus autosampler and 996-photodiode array detector (Waters). The column temperature was held at 30 °C, and samples in the autosampler were kept at 15 °C. Chromatographic separation was achieved with a flow rate of 1 mL min^−1^ and a linear gradient of 40%–90% (*v/v*) acetonitrile in 0.1% (*v*/*v*) formic acid over 5 min, then held for 0.5 min before the immediate change to 100% acetonitrile, and subsequently held for 5 min before returning to the original conditions of 40% acetonitrile for 10 min prior to the next injection [35]. Absorbance at 255 nm was used for the quantification of herbicides. Under these conditions, the retention times for each herbicide were: pyroxasulfone, 7.8 min; clethodim, 10.8 min; and trifluralin, 11.4 min. Positive identification of herbicides was made by comparing standard retention time and photodiode array (PDA) peak spectral analyses across 200–500 nm.

Glyphosate (1 mM in water) was derivatized with fluorenylmethoxycarbonyl (FMOC: Sigma-Aldrich) according to Ibáñez et al. [36], and 10 μL injections were separated at a flow rate of 1 mL min^−1^ on the same C_18_ column as above. The column temperature was held at 35 °C, and samples in the autosampler were kept at 15 °C. The mobile phase was a gradient of ammonium acetate (pH 5.02) and acetonitrile adapted from Ibáñez et al. [36], with initial conditions of 10% acetonitrile followed by the immediate change to 35% acetonitrile after 0.1 min, then held at 35% for 10 min before the immediate change back to 10% for column equilibration prior to the next injection. Derivatized glyphosate was detected with a 996-photodiode array detector (Waters) at 265 nm. Under these conditions, the retention time for glyphosate was 7.3 min. Positive identification of derivatized glyphosate was made by comparing standard retention time and PDA peak spectral analyses across 210–350 nm.

Subsequent HPLC analysis of the commercially formulated herbicides at the same concentration of active ingredient showed that there was no interference with the herbicide peaks from other compounds present in the formulations (data not shown). The detection limit was approximately 0.2 nmol per injection.

Stock solutions of each formulated herbicide were prepared by diluting in water to a concentration of 10 mM active ingredient. The solutions were stored in glass bottles at 4 °C in the dark, and aliquots were removed at regular intervals and stored at −20 °C for HPLC analysis of two technical replicates. The stability of each formulated herbicide, when dissolved in a 0.6% (*w*/*v*) agar matrix, was tested by preparing agar containing 2 μM clethodim, 400 μM glyphosate, 0.15 μM pyroxasulfone, or 20 μM trifluralin. Dishes of agar were prepared for storage at (i) ambient laboratory temperature (mean 21 °C) and light (mean 50 μmol m^−2^ s^−1^ during the day); (ii) ambient laboratory temperature in the dark (dishes wrapped in foil); or (iii) 4 °C in the dark, with the replicated experiment having three replicates per treatment. Pilot studies on samples stored in the 25/15 °C growth cabinet showed that there was no difference in herbicide degradation under ambient laboratory conditions compared to the growth cabinet (data not shown). The total mass of the agar in each dish was recorded, and then samples were taken using a 12 mm cork-borer at 0, 5, and 10 d after preparation. The mass of each sample was recorded, and the agar plugs were then immediately chopped into 1 mm^3^ pieces and extracted by vigorous agitation in 100% acetonitrile (clethodim, pyroxasulfone, and trifluralin) or in water (glyphosate). Samples were then centrifuged at 18,000 *g* for 10 min to pellet the agar. The supernatant volumes were recorded, and the trifluralin and glyphosate samples were immediately stored at −80 °C. The pyroxasulfone and clethodim samples were concentrated under a stream of air, partitioned against an equal volume of ethyl acetate to remove the water, evaporated to dryness, and then redissolved in 30–50 μL of 100% acetonitrile and stored at −80 °C until HPLC analysis was performed.

### 4.7. Statistical Analysis

Dose-response curves were constructed for each population, herbicide and test system (agar vs. pot test). Data from repeated experiments (completely randomized design) were pooled and subjected to non-linear regression analysis, using a log-logistic model with three parameters:Y = d/1 + exp b(log(x) − log(e))(1)
where Y corresponds to the survival response to herbicides for each population, d is the upper limit, b is the slope of the curve, x is the herbicide dose, and e is the dose causing 50% mortality response [37]. Thus, the lethal dose required to kill 50% of the population (LD_50_) and respective 95% confidence intervals were estimated for each herbicide and each population from the dose-response analysis. The resistance index (RI) was calculated as the ratio of the LD_50_ of the resistant population and standard susceptible population S.

Correlation analysis was conducted on plant survival percentage values obtained in 100 annual ryegrass populations, tested at the discriminating dose in agar tests vs. pot tests. Pearson’s correlation coefficient (r) and two-tailed *p*-values were estimated for pair-wise combinations of plant survival observed in agar and pots for each of the four herbicides tested in the study. The statistical analysis was conducted with GraphPad Prism (GraphPad Software, Inc., La Jolla, CA, USA). Herbicide stability and seed germination data were analysed using one-factor ANOVA and Welch’s t-test for pair-wise comparisons of treatments.

## Figures and Tables

**Figure 1 plants-10-01879-f001:**
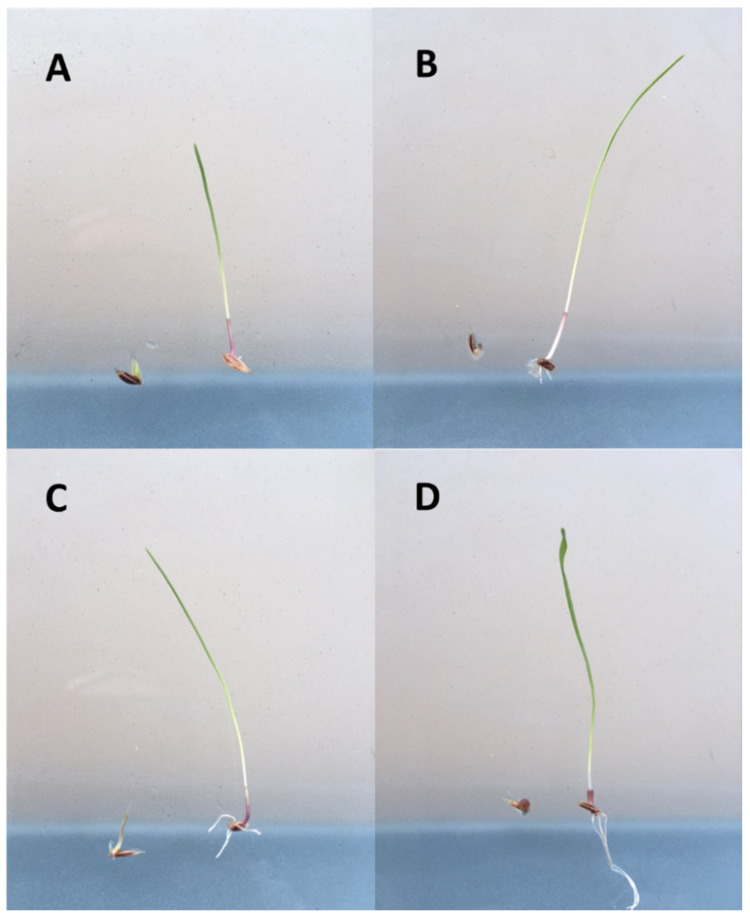
Inhibition of seedling growth of susceptible (**left**) and resistant (**right**) *Lolium rigidum* after 7 d when treated with a discriminating dose of (**A**) clethodim (1 µM), (**B**) glyphosate (400 µM), (**C**) pyroxasulfone (0.15 µM), and (**D**) trifluralin (25 µM).

**Figure 2 plants-10-01879-f002:**
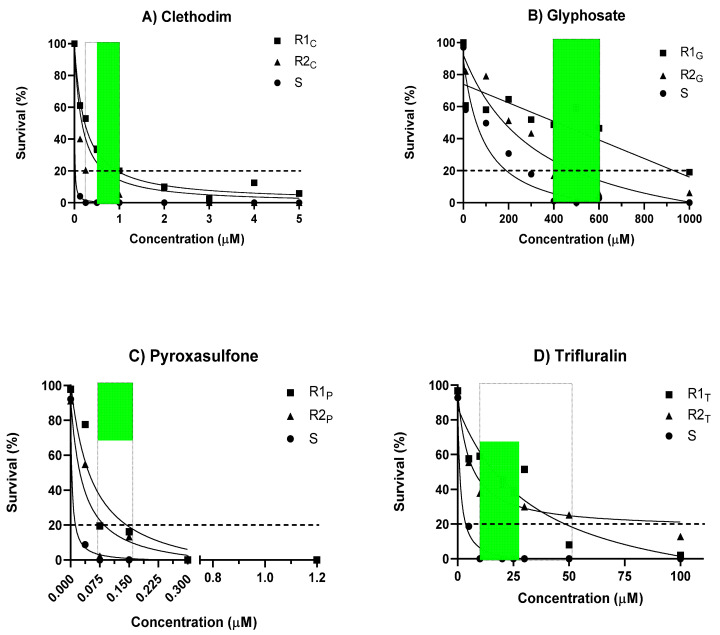
Dose-response curves (see text for regression equation) for seedlings of susceptible (S) and resistant (R) populations of *Lolium rigidum* germinated on agar containing increasing concentrations of (**A**) clethodim; (**B**) glyphosate; (**C**) pyroxasulfone; or (**D**) trifluralin (each data point is the average of two experimental runs with three replicates per concentration; the dotted line represents a discriminating threshold to identify a highly resistant population (>20% survival) while the shaded area defines the range of concentrations that could be used to detect herbicide-resistant phenotypes (0% survival of S vs. 2%–60% survival of R).

**Figure 3 plants-10-01879-f003:**
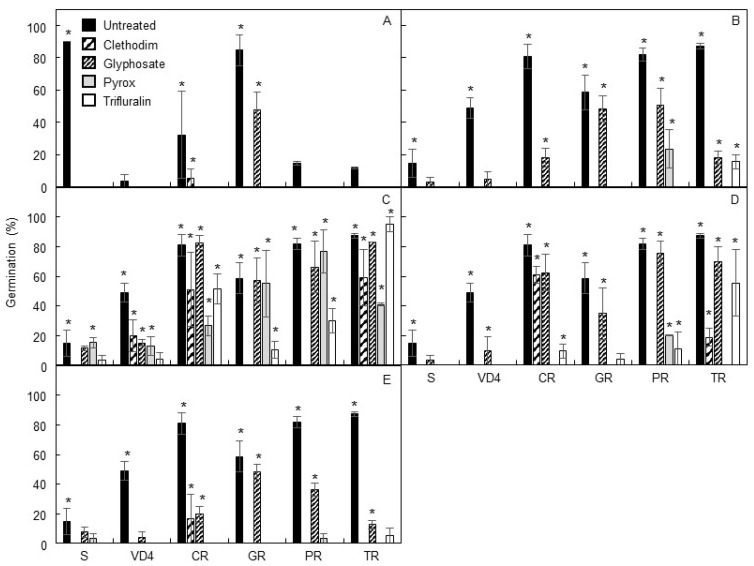
Apparent resistance levels (means ± SE, measured as the percentage of seeds germinating after 7 d and producing a coleoptile of ≥4 cm) of population S, non-dormant susceptible control; VD4, dormant susceptible control; CR, clethodim-resistant; GR, glyphosate-resistant; PR, pyroxasulfone-resistant; TR, trifluralin-resistant when placed on herbicide-containing (or untreated) agar under germination conditions of 25/15 °C with a 12 h photoperiod: (**A**) direct imbibition; (**B**) 21 d dark stratification in the absence of herbicides; (**C**–**E**) 21 d dark stratification on 0.1×, 0.25×, and 1× of the herbicide discriminating dose, respectively (2 μM clethodim, 400 μM glyphosate, 0.15 μM pyroxasulfone, or 20 μM trifluralin); herbicide treatments resulting in ≥20% germination (*p* < 0.05), indicating apparent resistance, are marked with an asterisk.

**Figure 4 plants-10-01879-f004:**
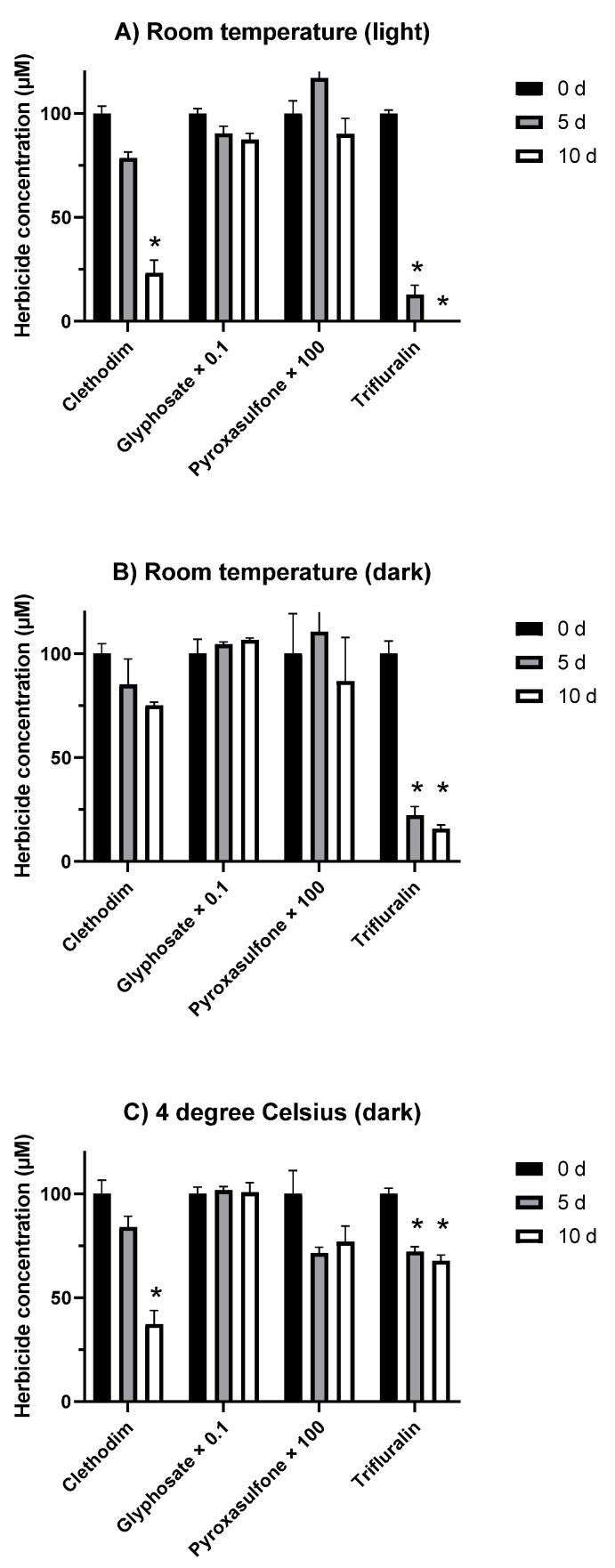
Stability of herbicides in agar (herbicides incorporated at concentrations of 2 μM clethodim, 400 μM glyphosate, 0.15 μM pyroxasulfone, or 20 μM trifluralin) stored at (**A**) room temperature under ambient light; (**B**) room temperature in the dark; or (**C**) 4 °C in the dark and monitored at 0, 5, and 10 d after the start of the experiment (values are means ± SE; asterisks (*) above bars denote herbicide doses that are significantly (*p* < 0.05) different from the corresponding starting concentration (0 d).

**Table 1 plants-10-01879-t001:** The lethal dose required to kill 50% (LD_50_) of individuals of each population in the agar and pot test. LD_50_ values (with 95% confidence intervals in parentheses) were calculated from a three-parameter non-linear regression model (see text). The resistance index (RI) was calculated based on the ratio of LD_50_ of the resistant (R) population to the susceptible (S) population. All estimated LD_50_ values of the R populations were significantly different from S (*p* < 0.05).

Herbicide	Population	Agar Test (µM)		Pot Test (g ha^−1^)	
		LD_50_ (95% CI)	RI	LD_50_ (95% CI) ^1^	RI
Clethodim	R_1C_	0.23 (0.13–0.43)	4.6	>750	>17
Clethodim	R_2C_	0.18 (0.05–0.76)	3.6	>750	>17
Clethodim	S	0.05 (0.004–0.007)	--	45 (21–68)	--
Glyphosate	R_1G_	370 (189–1110)	4.0	>1350	>7
Glyphosate	R_2G_	347 (164–770)	3.8	>1350	>7
Glyphosate	S	91 (38–184)	--	184 (143–225)	--
Pyroxasulfone	R_1P_	0.06 (0.03–0.095)	15	46 (26–66)	7.7
Pyroxasulfone	R_2P_	0.03 (0.015–0.06)	7.5	22 (14–30)	3.7
Pyroxasulfone	S	0.004 (0.0004–0.007)	--	6.0 (2.1–9.9)	--
Trifluralin	R_1T_	35 (6–391)	29	>720	>8
Trifluralin	R_2T_	5.4 (0.3–41)	4.5	203 (54–352)	2.1
Trifluralin	S	1.2 (0.6–1.8)	--	96 (80–112)	--

^1^ If an LD_50_ value could not be calculated due to >50% survival at the highest dose tested, the LD_50_ value was designated as greater than that dose.

**Table 2 plants-10-01879-t002:** Correlation coefficients (r) of herbicide survival response obtained in pot-based vs. agar-based studies by screening 100 populations of *Lolium rigidum* (ns, not significant; **, *p* < 0.01; ***, *p* < 0.001).

Agar					
		Clethodim	Glyphosate	Pyroxasulfone	Trifluralin
	Clethodim	0.426 ***			
Pots	Glyphosate	−0.034 ns	0.161 ns		
	Pyroxasulfone	0.017 ns	−0.092 ns	−0.098 ns	
	Trifluralin	0.076 ns	−0.023 ns	−0.211 ns	0.337 **

**Table 3 plants-10-01879-t003:** Populations of known resistance status used to optimize the agar test.

Population	Resistance Profile	Year Collected	Population Origin/Code	Reference
S	Susceptible	1985	VLR1	Neve and Powles [24]
VD4	Susceptible	2010 ^1^	Wongan Hills, WA	Goggin et al. [23]
R_1C_	Clethodim	2003	M1/25	Owen et al. [25]
R_2C_	Clethodim	2015	H3/3	Owen et al. [8]
CR	Clethodim	2018	WA4-18	R. Busi [unpublished data]
R_1G_	Glyphosate	2015	AFLR2	Yu et al. [26]
R_2G_	Glyphosate	2015	NLR70	Lorraine-Colwill et al. [27]
GR	Glyphosate	2018	WA124-18	R. Busi [unpublished data]
R_1P_	Pyroxasulfone	2009 ^1^	MR-P3 ^2^	Busi et al. [25]
R_2P_	Pyroxasulfone	2014 ^1^	MR-P6 ^2^	Busi and Powles [28]
PR	Pyroxasulfone	2019 ^1^	M3/54-P4 ^2^	R. Busi [unpublished data]
R_1T_	Trifluralin	1982	SLR31	Heap and Knight [29]
R_2T_	Trifluralin	2012	WA678-13	P. Boutsalis [unpublished data]
TR ^3^	Trifluralin	2018	WA3-18	R. Busi [unpublished data]

^1^ The year that population was selected for resistance/dormancy in the laboratory. ^2^ Low-dose selection with pyroxasulfone to obtain a pyroxasulfone-resistant progeny from parental field populations SLR31 and M3/54. ^3^ Low-level resistance to clethodim observed in pots (approx. 10% survival to 60 g clethodim ha^−1^).

## Data Availability

Not applicable.

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
