# Peer review of "Rapid On-Farm Testing of Resistance in Lolium rigidum to Key Pre- and Post-Emergence Herbicides"

_plants, 2021, doi:10.3390/plants10091879_

Round 1

Reviewer 1 Report

In the manuscript plants-1355769, Badano-Perez et al. present an interesting proposal for the identification of resistance to key PRE and POST-emergent herbicides in L. rigidum using a rapid agar germination test.

The initiative is very interesting, and although the methodology still has some technical limitations so that the test can be used in the field (farmers and consultants) related to the dormancy of seeds and degradation of herbicides such as trifluralin, which the authors have explored different methods to overcome them, it is an interesting investigation that could begin to adapt to other species and herbicides

In general, the experiments were well designed and conducted, and in this regard I have two observations: 1) the light intensity used in the experiments is too low to induce germination (30-60 μmol m-2 s-1), considering that under field conditions it is approximately 1000 µmol m-2 s-1); and 2) I am amazed at the high detection and stability of glyphosate in glass. From experience, this molecule has a very irregular behavior in glass materials, even when a solution is analyzed immediately after its preparation, so it is generally recommended to use plastic pipettes / vials for handling and storage of samples with glyphosate. I would like you to explain / justify in more detail the conditions of analysis and manipulation of glyphosate in the HPLC experiments.

For the rest, the manuscript is well written and organized and in general terms, we ask the authors to present the main results in the abstract (numerical references), and for the results described in section 2.2, present an item (table/figure/supplementary material), or cite adequately in which tables or figures the results in this section can be seen. Minor corrections mainly related to the order/way in which the figures are displayed are highlighted in the attached PDF.

Reviewer 2 Report

Of the numerous manuscript reviews I have done over the years, this manuscript stands out for its exceptional writing, methodology and significance to the discipline of Weed Science. I commend the authors for a well-designed, executed, and written paper. I have no corrections and do recommend the manuscript be accepted in its current form.

Reviewer 3 Report

Authors have very clearly stated the need of rapid testing in the current times. The materials and methods are nicely explained with appropriate references. A revision for minor spelling check is needed. Overall, a well written manuscript. A suggestion to the authors, if they can add cost of the kit to the growers. If not, its understandable that this paper is not about the economics. 

Question:

1) Was the pre-emergence (PRE) herbicide dose response also conducted in potting mix soil? Normally, for PRE assays field soil is used. If yes, it is not clearly mentioned.
